# In vivo detection of antisense HIV-1 transcripts in untreated and ART-treated individuals

Adam A Capoferri[1],*, Rachel Sklutuis[1],*, Toluleke O Famuyiwa[1], Sachi Pathak[1], Rui Li[2], Jason W Rausch[1], Brian T Luke[3], Rebecca Hoh[4], Steven G Deeks[4], John W Mellors[5], John M Coffin[6], Jennifer L Groebner[1], Fabio Romerio[2], Mary F Kearney[1]

Natural antisense transcripts (AST) are expressed in eukaryotes, prokaryotes, and viruses and can possess regulatory functions at the transcriptional and/or post-transcriptional levels. In vitro studies have shown that HIV-1 AST promote viral latency through epigenetic silencing of the proviral 5′ long terminal repeat. However, expression of AST in vivo has not been convincingly demonstrated. Here, we used single RNA template amplification and sequencing to demonstrate expression of AST in unstimulated PBMC collected from people with HIV-1 (PWH). Our results show that expression levels of AST could be higher during ART compared with untreated individuals and that clones of infected cells persisting under ART continue to express HIV AST. This study is the first to verify HIV-1 AST expression in vivo with sequencing, documenting AST presence without cellular activation and suggest its natural occurrence in PWH. These findings advance our understanding of HIV-1 persistence and underscore the need for larger studies to determine if targeting AST in viral reservoirs could lead to new approaches for the design of strategies towards achieving HIV remission without ART.

## Introduction

Antisense transcripts (AST) are RNA molecules transcribed from the opposite strand of a protein-coding gene that can have protein-coding and/or non-coding activities (Khorkova et al, 2014). AST have been identified in eukaryotes, prokaryotes, and viruses and have been shown to possess regulatory functions at both the transcriptional and post-transcriptional levels via multiple mechanisms (Li et al, 2021). AST have previously been documented to be encoded by several viruses that infect eukaryotes including members of the *Herpesviridae* (e.g., herpes simplex virus-1 and cytomegalovirus) and *Retroviridae* families (Larocca et al, 1989; Zwaagstra et al, 1989; Michael et al, 1994; Kondo et al, 1996; Rasmussen et al, 2010). One of the best characterized AST in a viral system is the *Hbz* gene in Human T-cell Leukemia Virus Type 1 (HTLV-1). The interplay between *Hbz* (antisense) and *Tax* (sense) RNA expression and their protein products is thought to modulate the regulation of cellular pathways that promote survival and proliferation of HTLV-1 infected cells, thereby influencing the progression into adult T cell leukemia/lymphoma or HTLV-1-associated myelopathy/tropical spastic paraparesis (Bangham et al, 2019).

R.H. Miller first provided evidence of an antisense gene (named *asp*) overlapping the *env* gene in the HIV-1 genome (Miller, 1988). Several in vitro studies have since demonstrated the presence of a Tat-independent negative sense promoter in the HIV-1 3′ long terminal repeat (LTR) (Michael et al, 1994; Bentley et al, 2004; Landry et al, 2007), which drives the expression of multiple AST (Kobayashi-Ishihara et al, 2012) with both protein coding (Gholizadeh et al, 2021) and non-coding functions (Ma et al, 2016). The presence of HIV-1 antisense protein (ASP)-specific antibodies in serum (Caetano et al, 2024) and cytotoxic CD8+ T-lymphocytes (CTLs) in blood samples (Bet et al, 2015) have been detected from people living with HIV (PWH), thus providing indirect evidence for the expression of ASP in vivo. Detection of ASP has been reported in eight chronically infected lymphoid and myeloid cell lines during latent and productive infection (Affram et al, 2019). HIV-1 AST has been shown to inhibit viral replication and to promote and maintain latency in stably expressing CD4+ T cell lines (Zapata et al, 2017; Kobayashi-Ishihara et al, 2018). Although in vitro studies suggest that AST function in viral latency, only a few reports have examined its expression in vivo. Zapata et al reported low levels of AST in 3 PWH on antiretroviral therapy (ART) (>2 yr with undetectable levels of viremia) using real-time PCR in unstimulated CD4+ T cells Zapata et al (2017) and Mancarella et al (2019) detected AST in CD4+ T cells collected from both untreated and ART-treated PWH after ex vivo

[1]HIV Dynamics and Replication Program, National Cancer Institute, Frederick, MD, USA  [2]Department of Molecular and Comparative Pathobiology, Johns Hopkins University School of Medicine, Baltimore, MD, USA  [3]Advanced Biomedical Computational Science, Frederick National Laboratories for Cancer Research, Frederick, MD, USA  [4]Department of Medicine, University of California, San Francisco, CA, USA  [5]Division of Infectious Diseases, Department of Medicine, University of Pittsburgh School of Medicine, Pittsburgh, PA, USA  [6]Department of Molecular Biology and Microbiology, Tufts University, Boston, MA, USA

Correspondence: adam.capoferri@nih.gov; kearneym@nih.gov
*Adam A Capoferri and Rachel Sklutuis contributed equally to this work

**Table 1.  Donor demographics**

| Group | Participant identifier (PID) | Cohort location | Age (years) | Sex[a] | Race/ethnicity[b] | Minimum duration of infection (years) | HIV plasma RNA copies/ml at sampling[c] | ART regimen at sampling[d] | Duration on ART at sampling |
|---|---|---|---|---|---|---|---|---|---|
| Donors on ART | 1079 | SCOPE, USCF | 64 | M | Latino | 17.0 | <50 | FTC/TDF, ETV | 12.8 yr |
| | 1683 | SCOPE, USCF | 45 | M | White | 8.0 | <50 | FTC/TDF, DRV/r | 5.4 yr |
| | 2669 | SCOPE, USCF | 50 | M | White | Unknown | <50 | ABC/3TC/DTG | 4.3 yr |
| | | | 51 | | | | <50 | | 5.5 yr |
| | | | 56 | | | | 112 | | 2 wk[e] |
| | | | 56 | | | | <50 | | 1 mo[e] |
| Untreated | 291 | University of Pittsburgh | 53 | M | Black | 9.0 | 184,243 | Not on ART | 0[f] |
| | 477 | University of Pittsburgh | 26 | F | Black | 5.0 | 139,845 | Not on ART | 0[f] |
| | 1508 | SCOPE, USCF | 53 | M | Black | 30.0 | 128 | Not on ART | 0[f] |
| | 1775 | SCOPE, USCF | 37 | M | White | 5.0 | 1,704 | Not on ART | 0[f] |
| | 3611 | SCOPE, USCF | 37 | M | Black | 4.0 | 275,685 | Not on ART | 0[f] |

[a]Sex: M, male and F, female.
[b]Race/ethnicity reported by participant.
[c]Level of plasma viremia was determined by either Abbott Real-time HIV-1 Assay (SCOPE, UCSF) or COBAS HIV-1 Test (University of Pittsburgh).
[d]ABC, abacavir; DTG, dolutegravir; DRV, darunavir; ETV, etravirine; FTC, emtricitabine; TDF, tenofovir disoproxil fumarate; /r, ritonavir-boosted; 3TC, lamivudine.
[e]After 5.5 yr on ART, the participant had an unexpected ART interruption for ~4 wk. They reinitiated ART with the first timepoint post-ART interruption at 2 wk with low but detectable HIV-1 plasma viremia. Then 1 mo post-ART interruption with plasma viremia suppressed.
[f]Donors were not on ART.

stimulation with anti-CD3/CD28. However, no studies have provided sequence evidence for expression of AST in unstimulated cells collected from PWH. Therefore, we set out to determine if we could measure and genetically characterize HIV-1 AST in unmodified PBMC collected from donors on and off ART, and to compare their expression levels to that of HIV-1 sense transcripts in the same donors. Investigating expression of HIV-1 AST in vivo may contribute to our understanding of HIV-1 persistence and reveal new targets for controlling HIV-1 expression without ART.

## Results

### Participants and samples

To determine the levels of AST during chronic infection, PBMC were collected from 3 PWH on ART (McManus et al, 2019) and 5 PWH who were not on ART (Table 1) and who were enrolled either at University of California of San Francisco under the SCOPE trial (clinical trial # NCT00187512) or at University of Pittsburgh under the Optimization of Immunologic and Virologic Assays for HIV Trial (IRB# STUDY20040215). All donors provided written informed consent for the study. For one donor (PID 2669), samples collected at four longitudinal timepoints before and after an ART interruption and re-initiation were available.

Most participants were male (n = 7/8). Race/ethnicity as reported by donors was Black (n = 4/8), White (n = 3/8), and Latino (n = 1/8). Donors were diagnosed with HIV-1 subtype B with a minimum duration of infection median of 8 yr [IQR 5–13 yr] prior to sample collection; however, this information was unknown for one participant (PID 2669), who was on study more than 5 yr. Donors not on ART (untreated) were either ART naïve or not currently on an ART regimen due to a planned or unplanned interruption, and their plasma viremia levels were detectable (median: 139,945 range: 128–275,685 HIV-1 RNA copies/ml). All but one of the samples collected from PWH on ART had undetectable levels of viremia (<50 HIV-1 RNA copies/ml) with durations of treatment ranging from 2 wk to 12.8 yr. The one sample with detectable viremia on ART (112 HIV-1 RNA copies/ml) was collected from PID 2669 2 wk after ART re-initiation after an unplanned 4-wk treatment interruption. Levels of plasma viremia were measured as HIV-1 RNA copies/ml by either Abbott real-time assay or COBAS HIV-1 test.

### HIV-1 AST were detected in ART-treated and untreated PWH, with potentially higher levels in samples collected during ART

Our approach for detecting and measuring levels of HIV-1 AST were modified from previous methods and are described in the Materials

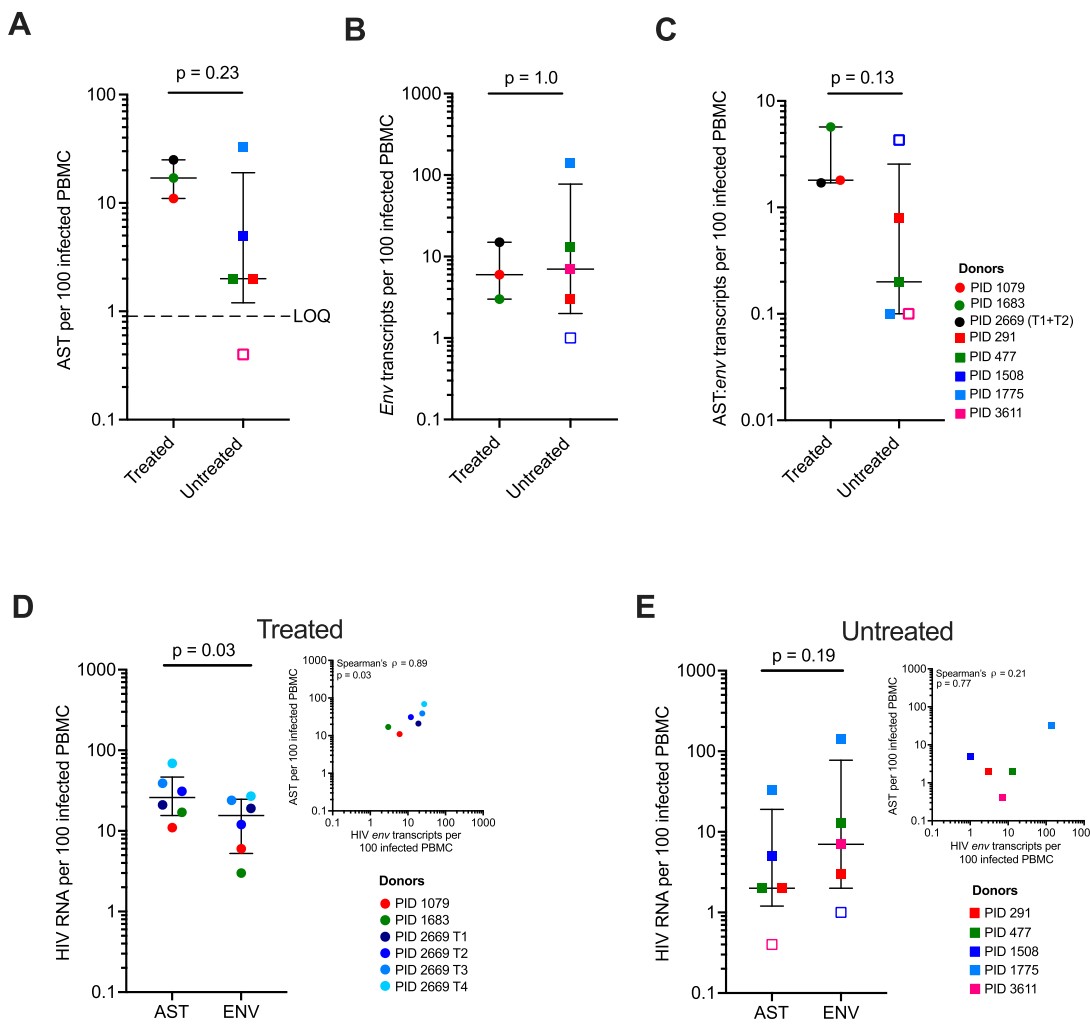

**Figure 1. Levels of HIV-1 antisense transcripts (AST) and *env* transcripts in PWH.**
**(A)** Levels of HIV-1 AST in ART-treated and untreated PWH, *P*-value determined with Wilcoxon rank-sum test. Open shape indicates a sample where AST was detected but was below the assay cut-off (details in the Materials and Methods section). **(B)** Detection of HIV *env* transcripts in ART-treated and untreated PWH, *P*-value determine with Wilcoxon rank-sum test. **(C)** Ratio of AST to *env* transcripts normalized to 100 infected PBMC, *P*-value determined with Wilcoxon rank-sum test. **(D)** Detection of AST versus *env* transcripts in ART-treated PWH, *P*-value determined with Wilcoxon signed-rank paired test. Inset plot for AST and *env* transcript with Spearman's correlation. **(E)** Detection of AST versus *env* transcripts in untreated PWH, *P*-value determined with Wilcoxon signed-rank paired test. Inset plot for AST and *env* transcripts with Spearman's correlation. PWH on ART (circles) and PWH not on ART (squares), each color represents an individual, and PID 2669 has multiple shades of one color to indicate longitudinal sampling and the aggregate timepoints 1 and 2 is a black circle. Open shapes indicate a sample were either AST was below limit of quantification or *env* transcripts was not detected.
Source data are available for this figure.

and Methods section (Wiegand et al, 2017; Zapata et al, 2017; Capoferri et al, 2024). Briefly, to measure levels of HIV-1 AST in vivo, we extracted total cell-associated RNA from unmodified donor PBMC with known estimated numbers of HIV-1 infected cells (Hong et al, 2016), synthesized cDNA using participant-specific exogenous oligo-tagged gene-specific primers targeting AST in the *env* coding region (Zapata et al, 2017), and quantified the number of AST in each sample with participant-specific anti-*env* primers and probes in a digital PCR format (primer/probe sequences in Tables S1, S2, S3, S4, and S5). Prior to testing the donor PBMC, we optimized the AST assay on antisense RNA in the ACH-2 cell line and found these cells to express a median of 41 AST per 100 infected ACH-2 cells (see the Materials and Methods section). Extensive controls were performed

to ensure complete degradation of HIV-1 DNA (Wiegand et al, 2017; Capoferri et al, 2024) and to determine the cut-off for HIV-1 AST (details in the Materials and Methods section). Equal numbers of no reverse transcriptase wells (negative controls) and experimental wells were included on each PCR plate to ensure no HIV-1 DNA contamination.

AST were detected in 7/8 donors (Fig 1A and Table S6). Because PID 2669 was overrepresented in the ART-treated group, which violates the assumption that observations within each group are independent of one another, we performed the statistical analysis by aggregating the AST in the two timepoints prior to the ART interruption for this donor. With this aggregate, the levels of AST in the ART-treated donors were a median of 17 AST/100 infected PBMC

[IQR 11–25] and levels in untreated donors were a median of ≤2 AST/100 infected PBMC [IQR 1–19] ($P$ = 0.23, Wilcoxon rank-sum test). However, when all four timepoints from PID 2669 are included, there is a median of 26 AST/100 infected PBMC in the donors on ART versus ≤2 AST/100 infected PBMC in the untreated donors, encouraging trials with more power to investigate the possibility that AST levels may be higher in donors on ART compared with untreated PWH. In PID 3611 (untreated), only 1 AST molecule was detected in 244 infected cells, which was below the cut-off for the assay (details for cut-off in the Materials and Methods section), indicating that the AST in this donor sample was ≤0.4 AST/100 infected PBMC. In PID 2669 (treated), AST were detected in samples collected at all four time-points: 4.3 and 5.5 yr after ART initiation and 2 and 4 wk after ART re-initiation following a brief treatment interruption. Samples after ART re-initiation had higher levels of AST than samples collected prior to ART interruption (49% of 133 infected PBMC versus 25% of 155 infected PBMC) ($P$ = 3.4 × 10$^{-9}$, binomial test).

### HIV-1 AST were detected at modestly higher levels than HIV-1 *env* transcripts in samples on ART

We measured the levels of *env* transcripts in the same sub-genomic region targeted for AST in both treated and untreated participants. *Env* transcripts were detected in 7/8 donors (Fig 1B and Table S7). Levels of *env* transcripts (aggregate of unspliced and partially spliced) were quantified in replicate aliquots of PBMC as those used for AST measurements. *Env* transcript levels were not significantly different in the treated and untreated participants (median 6 *env*/100 infected PBMC [IQR 3–15] versus median ≤7 *env*/100 infected PBMC [IQR 2–78]) ($P$ = 1.0, Wilcoxon rank-sum test) (Fig 1B) or 16 versus ≤7 *env*/100 infected PBMC when including multiple timepoints from PID 2669. Of note, in PID 1508, no *env* transcripts were detected (estimated 86 infected PBMC assayed). We found the ratio of AST:*env* transcripts to be higher in the treated (median 1.8 AST:*env*) versus untreated (median 0.2 AST:*env*) group, but the difference was not significant ($P$ = 0.13, Wilcoxon rank-sum test) (Fig 1C).

AST levels in the treated group were slightly higher than *env* transcripts levels in the same samples (median 26 AST/100 infected PBMC [IQR 16–47] versus median 16 *env*/100 infected PBMC [IQR 5–25]) ($P$ = 0.03, Wilcoxon signed-rank paired test), resulting in a significant positive association between AST and *env* transcripts, albeit measurements for AST and *env* transcripts were performed on different cells since cDNA cannot be synthesized in both directions on RNA derived from the same cell (Spearman's $\rho$ = 0.89, $P$ = 0.03) (Fig 1D). In contrast, in the untreated donors, levels of AST were not significantly higher than *env* transcripts (median ≤2 AST/100 infected PBMC [IQR 1–19] versus median ≤7 *env*/100 infected PBMC [IQR 2–78]) ($P$ = 0.19, Wilcoxon signed-rank paired test) and no significant correlation was found (Spearman's $\rho$ = 0.21, $P$ = 0.77) (Fig 1E).

### Long fragments of HIV-1 AST were amplified in single infected cells from donors on ART

Because of the higher levels of AST in the donors on ART, we were able to amplify longer fragments in this subset. We amplified 1.7-kb fragments of AST spanning from the negative sense promoter in the

3′ LTR to anti-*env* (referred to here as "long AST"). Using a modified version of the CARD-SGS assay (Wiegand et al, 2017; Capoferri et al, 2024), we used the sequence data to estimate the fraction of infected cells with "long AST" and the levels of AST in single infected cells (details in the Materials and Methods section) (Table 2). We found that a median of 4.1% [IQR 1.6–5.2%] of infected PBMC had detectable levels of "long AST" with a median of 1.0 copies/cell [IQR 1.0–1.0] at a given point in time, indicating that the frequency of detection of "long AST" was lower than the short AST fragments detected by the digital PCR approach (above) which was consistent with prior findings that detection of shorter fragments sometimes had a greater level of sensitivity (Wiegand et al, 2017). In PID 2669, we found no significant difference in the fraction of cells with "long AST" or the levels of AST in single infected cells across the four time points (Kruskal-Wallis test, H(3) = 3.02, $P$ = 0.39). In addition, we found no significant difference when we aggregated the samples prior to ART interruption and after ART-reinitiation ($P$ = 0.83, Mann-Whitney $U$ test).

### Phylogenetic analysis of AST reveals that their expression originates from a diverse population of proviruses

Having successfully amplified the 1.7-kb segments of HIV-1 AST in the donors on ART by RT–PCR, we sequenced the resulting PCR products and performed phylogenetic analyses to compare the genetics of the AST to the proviruses in the same population of infected cells (Fig 2). Standard HIV-1 *env* DNA single-genome sequencing using PBMC from all three donors and lymph node mononuclear cells (LNMC) from two of the donors was performed previously (McManus et al, 2019) and the data used here as a reference. Neighbor-joining trees were reconstructed using HIV-1 *env* DNA from PBMC and LNMC and ~1.0-kb of the AST in the same genetic region. Trees were rooted on the HIV-1 subtype B consensus sequence. Symbols on the trees show *env* DNA from PBMC (black triangles), *env* DNA from LNMC (blue triangles), and AST (multi-colored squares where each color is obtained from a different aliquot of PBMC). AST matching *env* DNA in PBMC or LNMC are indicated with black arrows; AST from infected probable T-cell clones are indicated with blue arrows. In PID 1079, a cell expressing high levels of AST is indicated with a red arrow. The genetic diversity of AST was measured by average pairwise distance (APD) with predicted hypermutant sequences removed prior to the calculation (Rose & Korber, 2000).

We detected a high genetic diversity of AST (ranging from 0.7–2.5%) matching a diverse population of proviruses in both PBMC and in LNMC (1.1–2.1%), indicating that a wide variety of proviruses can express AST (Fig 2). In one aliquot of PBMC from PID 1079, we found a rake of 30 identical AST, suggesting that they may have originated from the same infected cell (red arrow, Fig 2A). Further, we found identical AST across multiple aliquots of infected PBMC in all three donors, indicating that these AST may be present in infected T-cell clones (blue arrows). In PID 1683, we also found identical PBMC AST matching proviruses in both PBMC and LNMC (Fig 2B). Whereas we did not directly assess the presence of AST in LNMC due to limited sampling with fine needle aspirates, these data suggest that AST may be expressed in cell clones that are present in tissues as well as in blood. AST were highly diverse in all four samples collected from PID 2669: T1 (2.9% APD), T2 (2.4%), T3 (2.6%), and T4 (2.4%). We also

**Table 2. Fraction of infected cells with HIV-1 "long AST" in donors on ART.Source data are available for this table.**

| Participant identifier (PID) | Duration on ART at sampling | Estimated number of infected cells assayed | Number of HIV-1 AST sequences obtained | Estimated number of infected cells with HIV-1 AST[a] | Estimate % of infected cells with HIV-1 AST | Median number of HIV-1 AST copies per cell (range)[b] |
|---|---|---|---|---|---|---|
| 1079 | 12.8 yr | 720 | 43 | 12 | 1.7 | 1.0 (1–30) |
| 1683 | 5.4 yr | 1,880 | 22 | 20 | 1.1 | 1.0 (1–2) |
| 2669 | 4.3 yr | 1,440 | 67 | 63 | 4.4 | 1.0 (1–3) |
| | 5.5 yr | 1,040 | 54 | 52 | 5.0 | 1.0 (1–2) |
| | ART interruption | | | | | |
| | 2 wk[c] | 2,000 | 86 | 74 | 3.7 | 1.0 (1–3) |
| | 1 mo | 720 | 41 | 40 | 5.6 | 1.0 (1–2) |
| | **Median** | **1,240** | **49** | **46** | **4.1** | **1.0** |
| | **IQR** | **720–1,910** | **36–72** | **18–66** | **1.6–5.2** | **1.0–1.0** |

"long AST" = 1.7 kb from 3′ LTR to *env*.
[a]Assumes AST with identical sequences are produced in the same single infected cell.
[b]Cells without HIV-1 AST were excluded.
[c]Not fully suppressed after 2 wk on ART following 4-wk unplanned ART interruption (ATI). PID 2669 was on ART for 5.5 yr prior to an ATI. Samples were collected at 4.3 and 5.5 yr prior to the ATI and at 2 wk and 1 mo after ART re-initiation after the ATI.

identified 11 likely T-cell clones that contained some cells with AST (blue arrows). Eight of the 11 clones were found to persist across multiple timepoints, mostly either before or after the ART interruption, and only very rarely persisting both before and after the interruption, suggesting that ART interruption may influence the populations of T cells that express HIV-1 AST (Fig 2C).

**AST can be detected in *gag*, *pol*, and *env* coding regions in vivo**

In vitro studies have detected AST of varying lengths, including "full-length" (i.e., from the U3/R of the 3′ LTR to *gag*) (Michael et al, 1994; Landry et al, 2007; Kobayashi-Ishihara et al, 2012) (Fig S1A). We asked if AST could be detected, not only in *env*, but in other HIV genomic regions in vivo. We measured AST in *gag* (HXB2: 764–2,281) (Myers et al, 1995), *gag*/*pol* (HXB2: 1,849–3,500) (Wiegand et al, 2017) and *pol*/*vif* (HXB2: 3,996–5,270) (Swanson et al, 2003) in one donor sample (PID 2669 Timepoint #4) (Fig 3A). Cell-associated RNA from three aliquots of ~90 infected PBMC was extracted and cDNA targeting the antisense strand of *gag*, *gag*/*pol*, and *pol*/*vif* was synthesized using exogenous oligo-tagged gene-specific primers, followed by endpoint PCR amplification and Sanger sequencing (primers in Table S1). We detected AST in all genomic regions assayed. However, we found lower levels of genetic diversity for AST in *gag*, *gag*/*pol*, and *pol*/*vif* regions compared with AST in *env* at the same time points (1.5% versus 3.2%) (Fig 3B). Comparable to levels of "long AST" in *env* (1.7 kb described above), levels of AST in *gag*, *gag*/*pol*, and *pol*/*vif* were 1.0, 1.1, and 1.0 copies/cell, respectively. Although the sub-genomic regions cannot be genetically linked, these findings suggest that AST may span the entire HIV-1 genome in some infected cells.

# Discussion

In this study, we used established quantitative PCR methods (Zapata et al, 2017), extensive positive and negative controls, and

sequencing to demonstrate the expression of HIV-1 antisense RNA in people living with HIV (PWH) who are either untreated or are on long or short-term ART. Prior to our study, HIV-1 antisense RNA had been shown to promote and maintain viral latency in stably expressing cell lines by recruiting the enhancer of zeste homolog 2 (EZH2), a core component of the polycomb repressive complex 2 (PRC2), to the HIV-1 5′ LTR (Zapata et al, 2017; Kobayashi-Ishihara et al, 2018). Recruitment of EZH2 catalyzes trimethylation of lysine 27 on histone H3 (H3K27me3), a suppressive epigenetic mark that promotes nucleosome assembly and suppression of viral transcription. HIV-1 AST are inefficiently polyadenylated and predominately retained in the nucleus to act as a lncRNA (Ma et al, 2021). Although HIV-1 AST have been shown to inhibit viral replication and promote the establishment and maintenance of latency in vitro (Saayman et al, 2014; Zapata et al, 2017), studies investigating AST in vivo have been limited. Therefore, we sought to determine if AST are expressed in vivo in untreated and/or ART-treated PWH, to quantify their levels in bulk and in single infected cells, and to characterize their genetics. To achieve this, we used a digital PCR assay for the detection and quantification of AST in unstimulated PBMC and we used a modified version of CARD-SGS (Wiegand et al, 2017; Capoferri et al, 2024) to measure the fraction of infected cells with longer fragments of AST (1.7 kb from 3′ LTR to *env*), the levels of AST in single infected cells, and the genetic diversity of AST in the unstimulated PBMC populations. Although methods for the CARD-SGS assay have been published previously (Wiegand et al, 2017; Capoferri et al, 2024), we included details in the Materials and Methods section.

We detected HIV-1 AST in 7/8 donors independent of treatment status. In additionally, we found 8.5-fold higher levels of AST in donors on ART compared with untreated donors, and 13-fold higher levels when including longitudinal samples from one donor, encouraging larger studies to investigate a possible role of AST in the maintenance of viral latency (Saayman et al, 2014; Zapata et al, 2017). An alternative interpretation could be that, in untreated donors, there are higher levels of transcription from the 5′ LTR. RNA

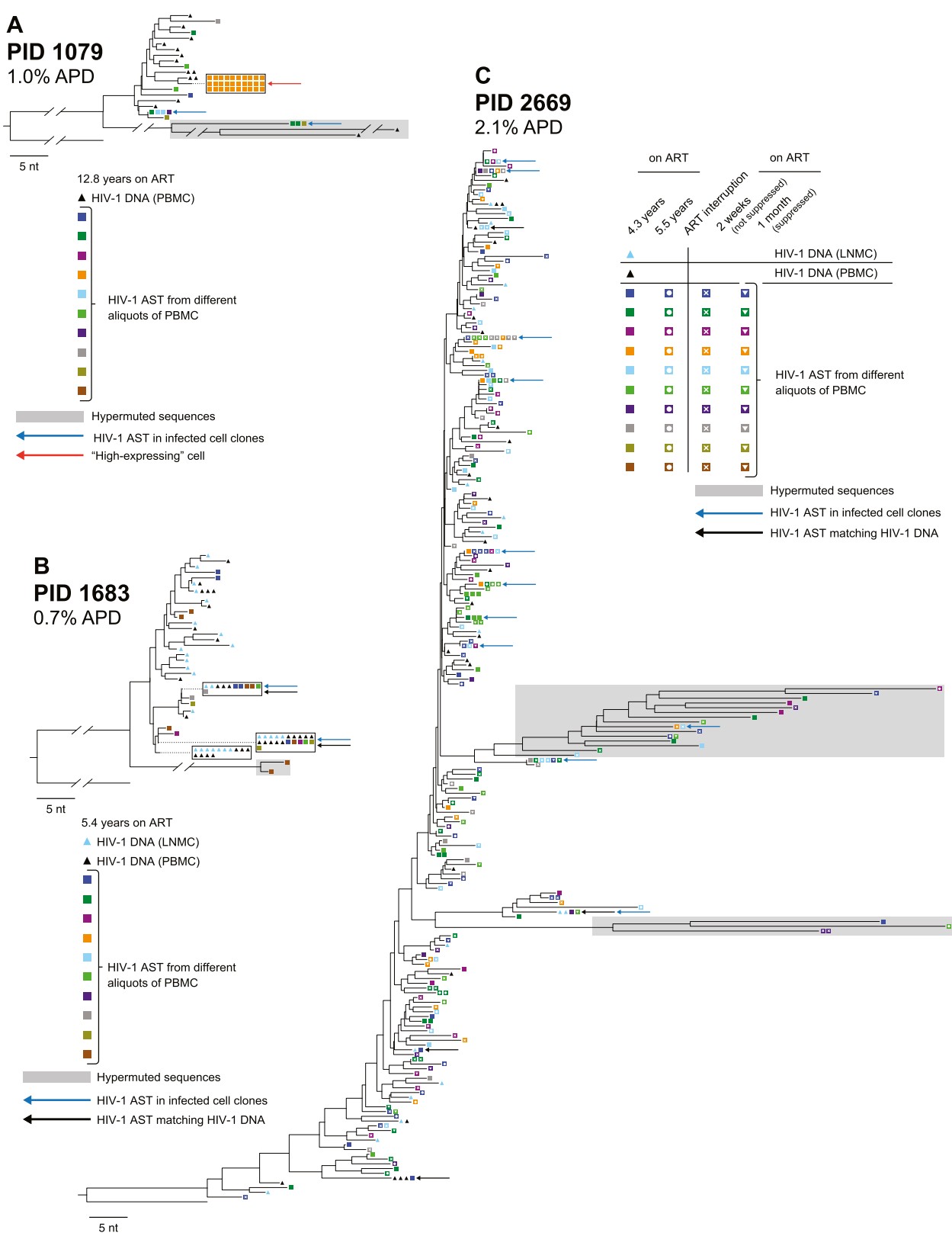

**Figure 2. Distance trees of antisense transcripts (AST) in donors on ART.**
**(A, B, C)** HIV-1 AST sequenced using the modified CARD-SGS assay for (A) PID 1079, (B) PID 1683, and (C) PID 2669 (see the Materials and Methods section, Fig S1). Percent average pairwise distance was calculated without hypermutants. HIV-1 AST sequences were aligned with the HIV-1 DNA *env* sequences (previously reported in McManus et al [2019]). Black triangles show proviruses from PBMC, light blue triangles show proviruses from lymph node mononuclear cells, and squares are the intracellular AST

polymerase collision (i.e., transcriptional interference, either "sitting duck" or "roadblock") or RNA:RNA hybrid formation between 5′ LTR-driven and 3′ LTR-driven transcription may suppress the expression of AST. We also detected HIV-1 *env* transcripts in 7/8 donors independent of treatment status. We found similar levels of *env* transcripts in ART-treated versus untreated individuals when normalizing for the number of infected cells analyzed, consistent with our prior studies showing that only about 10–20% of infected cells express HIV RNA at a given point in time irrespective of ART status (Wiegand et al, 2017; McManus et al, 2019; Capoferri et al, 2024). Interestingly, there was a trend for participants on ART to have higher AST-to-*env* ratios than untreated participants when all timepoints were included, which may reflect the ongoing rounds of viral replication in untreated donors.

Except for one donor on ART (PID 1079) where we identified one cell that may have had about 30 copies of AST, the level of AST in single cells was very low, with a median of 1.0 copy/cell (range 1–30 copies/cell). The low levels of HIV-1 AST are consistent with those reported for other eukaryotic AST. In about 25% of protein-coding genes, AST are expressed at ~1 or a few copies per cell at any given time (Djebali et al, 2012). In contrast, some AST can be expressed at very high levels, as observed for human *MALAT1* at about 150 TPM (GTEx Consortium et al, 2017). This range of antisense lncRNA expression has been well documented across cellular and tissue types in the Functional Annotation of the Mammalian Genome (FANTOM) (FANTOM Consortium and the RIKEN PMI and CLST DGT et al, 2014; Hon et al, 2017; Ramilowski et al, 2020), the Genotype-Tissue Expression (GTEx) consortium (GTEx Consortium et al, 2017), the Encyclopedia of DNA Elements (ENCODE) project (Djebali et al, 2012), and the Long non-coding RNA Knowledgebase (Seifuddin et al, 2020). Furthermore, the levels of HIV-1 AST that we report here are consistent with another retrovirus, HTLV-1, where a study reported 1–2 copies of the AST (*Hbz*) per HTLV-1+ cell, and a large fraction (20–70%) of the HTLV-1+ cells containing no detectable copies at any given time (Miura et al, 2019).

In the three donors on ART, we amplified and sequenced a 1.7-kb fragment of HIV-1 AST overlapping *env* from about 80 infected cells per sample. We found high genetic diversity of the AST including matches to proviruses in peripheral blood and lymph nodes. In some instances, the AST matched both PBMC and LNMC proviral DNA, suggesting that at least some cells in infected CD4⁺ T-cell clones can express HIV-1 AST. In the one donor on ART with samples from multiple timepoints (PID 2669), we identified probable T-cell clones with detectable levels of AST that persisted over time, and one that matched proviral DNA found in both PBMC and LNMC. Interestingly, although identical AST were found across the timepoints before the ART interruption and across the timepoints after the ART interruption and re-initiation, only rarely were identical AST detected both before and after the interruption, driving the hypothesis that treatment interruption may influence the populations of T-cell clones that express HIV-1 AST. Larger future studies are

needed to test this hypothesis. We also observed elevated AST levels in the samples collected within weeks after ART re-initiation relative to samples collected on longer term ART (4–5 yr on ART). While it is tempting to imagine that AST levels are influenced by ART duration and interruptions, it is not possible with our small sample size to know if these differences are a result of treatment status or only reflect the natural variation of AST expression in vivo. Future studies are needed to determine if AST levels are different in donors on and off ART, in donors with varying durations of treatment, and in donors who experience ART interruptions versus those who do not. Here, we can only definitively conclude that HIV-1 AST is expressed in donors on and off ART, is present in a small fraction of infected cells at a given point in time, is typically expressed at low levels, is derived from a highly diverse population of proviruses, and can be detected in probable infected T-cell clones in both blood and tissues. These findings set the foundation for larger studies that will determine the possible association of HIV-1 AST expression with treatment status, viral setpoints, and viral latency and reactivation.

Previous in vitro studies have reported multiple HIV-1 AST species using Northern blot and 5′ Rapid Amplification of cDNA Ends (5′ RACE): class I (10 kb) (Kobayashi-Ishihara et al, 2012), II (5.5 kb) (Landry et al, 2007), III-iii (3 kb) (Kobayashi-Ishihara et al, 2012), and IV-ii (2 kb) (Michael et al, 1994) (Fig S1A). Our detection of HIV-1 AST in genes other than in *gag*, *gag/pol*, and *pol/vif* at levels comparable to those we observed in *env*, together with the work by Kobayashi-Ishihara et al (2012), suggest that we detected numerous class I transcripts, which are full-length, expressed at low levels, and not typically polyadenylated. Relatedly, it is important to note that neither our digital PCR nor our single-genome sequencing assays targeting *env* is informative as to which class the identified AST transcripts belong.

There are additional limitations of this study to consider. Detection of higher levels of "shorter" AST fragments versus "long AST" may be due to degradation of RNA within cells (biological) and/or varying cDNA or PCR efficiencies (technical). Due to the fragility and sparsity of HIV-1 AST, we almost certainly underestimate their frequency. We cannot rule out that HIV-1 AST are short-lived and may not always be detectable. Similarly, there is a potential loss of AST from inefficient reverse transcription due to the thermal stability of secondary and tertiary RNA structures (although the high temperature denaturing step leading into cDNA synthesis is designed to help mitigate such effects), or during the sodium acetate/ethanol precipitation of the antisense cDNA. These recovery limitations are supported by the controls demonstrating recoveries of only ~50% for AST spiked samples (see the Materials and Methods section). Detecting a single copy of HIV-1 AST naturally presents challenges, particularly in determining if the molecules are due to expression by an HIV-1 or host promoter. Read-through generation of HIV-1 AST can occur when the provirus integrates opposite to a host gene promoter, allowing transcription into the HIV-1 3′ LTR to produce AST. Previous research identifying the

from PBMC with each color representing a different aliquot of PBMC. Blue arrows indicate identical AST sequences found in >1 aliquot of PBMC, black arrows indicate identical AST sequences that matched either PBMC or LMNC HIV-1 DNA, and the red arrow indicates a "high AST-expressing" cell. Predicted hypermutant sequences are shaded in gray. Trees are rooted on consensus subtype B *env*.

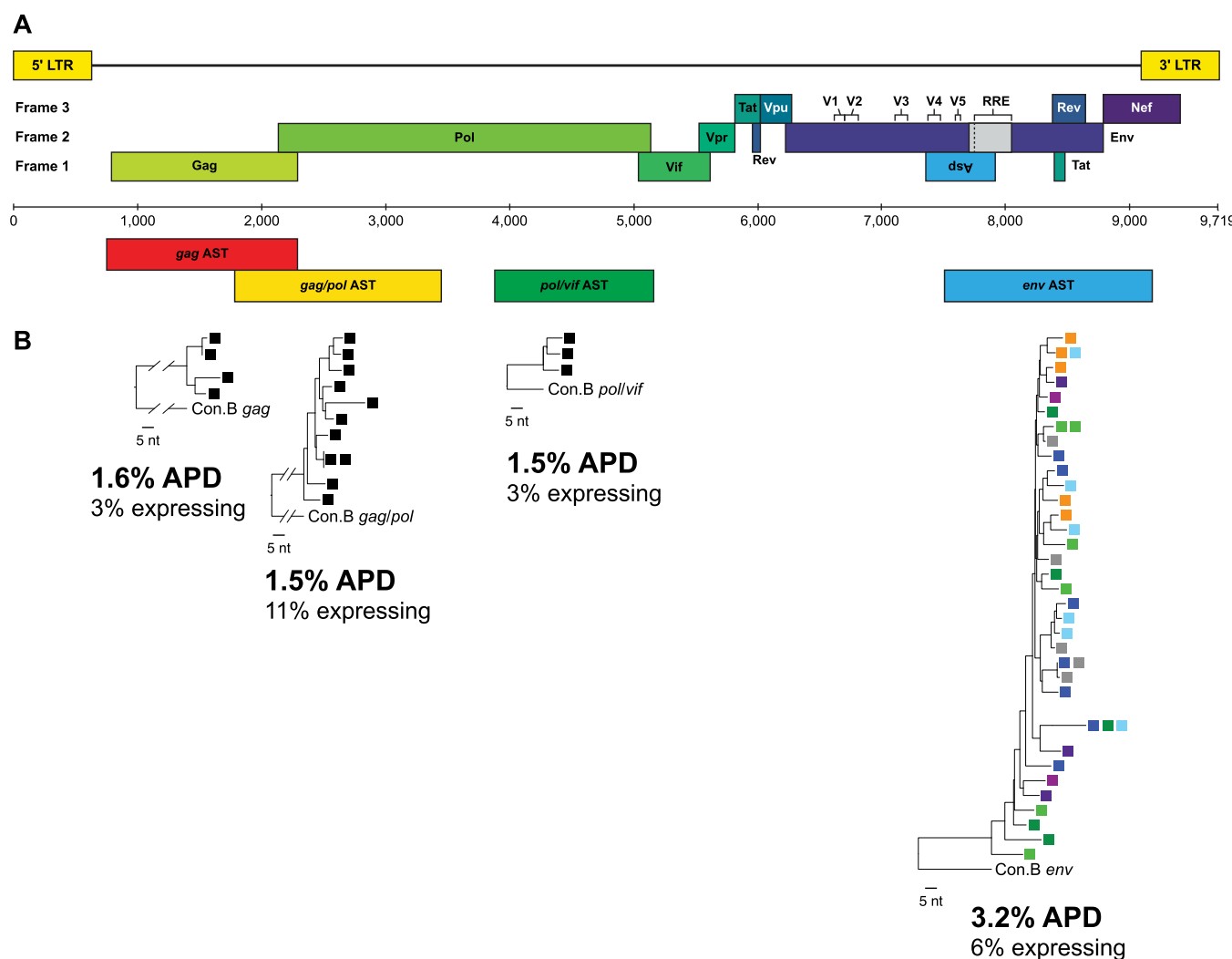

**Figure 3. Detection of antisense transcripts (AST) along the proviral genome.**
**(A)** HIV-1 genome map indicating sub-genomic regions for AST amplification. **(B)** Modified AST single-genome sequencing using exogenous oligo-tagged primers to synthesize cDNA and to PCR amplify AST along the genome. RNA from aliquots of 90 infected PBMC were extracted from PID 2669 Timepoint #4 for each sub-genomic region (anti-*gag*, *gag/pol*, *pol/vif*) and denoted by a single colored square. p-distance neighbor-joining phylogenetic reconstruction of positive amplicons were generated and percent average pairwise distance and fraction of expressing cells are reported. The AST tree for anti-*env* is extracted from Fig 2C, where up to 10 aliquots of PBMC were used. Each color square represents a different aliquot of PBMC from Timepoint #4.

transcription start site of HIV-1 AST in vitro used infected cell lines and 5′ RACE (Landry et al, 2007; Kobayashi-Ishihara et al, 2012). We attempted 5′ RACE on our donor samples to determine if the HIV-1 AST originated from an HIV-1 or host promoter at a single RNA molecule level, but without success due to limitations in assay sensitivity. More sensitive 5′ RACE technologies are needed to determine the transcription start site of single RNA molecules in vivo, including HIV-1 AST. However, our finding that there are differences in levels of AST across different cells of a T cell clone may favor expression from the HIV 3′ LTR promoter.

This study is the first to show, with sequence confirmation, the expression of HIV-1 AST in PWH. Larger studies are needed to confirm our observation of potentially higher levels of AST in PWH on ART versus PWH who are not on ART, and to determine if HIV-1 AST in vivo are driven by viral or host promoters. This study,

together with those showing the role of HIV-1 AST in inducing and maintaining HIV-1 latency in vitro and ex vivo (Li et al, 2025), highlights a previously underexplored potential determinant of HIV-1 persistence both before and during ART, and may lead to new directions for the development of approaches to controlling HIV-1 viremia without ART.

# Materials and Methods

### Participant cohorts, sample collection, and study approval

PBMC were collected from PWH on ART who were enrolled at the University of California of San Francisco in the SCOPE trial (clinical

trial # NCT00187512). PBMC were also collected from PWH who were not currently on ART and were enrolled at the University of Pittsburgh in the Optimization of Immunologic and Virologic Assays for HIV Trial (# STUDY20040215; IRB). The studies were approved by the University of California San Francisco Institutional Review Board and the University of Pittsburgh Institutional Review Board. All donors provided written informed consent for HIV-1 quantification and sequencing. PBMC were separated using Ficoll, resuspended in FBS with 10% DMSO, and stored in liquid nitrogen ($LN_2$) until testing.

## Overview of HIV-1 AST quantification and sequencing approaches

HIV-1 AST quantification and sequencing assays were performed by modifying our cell-associated RNA and DNA single-genome sequencing assay (CARD-SGS) (Wiegand et al, 2017; Capoferri et al, 2024) (detailed methods below). Briefly, total nucleic acid was extracted from serially diluted aliquots of PBMC collected from PWH, the DNA was digested, and AST cDNA synthesized using a gene-specific primer in the opposite orientation of the *env* coding region (Fig S1B). The cDNA from each aliquot was spread across 96-well PCR plates for amplification (~50 bp) and probe detection to determine the dilution that yielded <30% positive PCR products, indicating that cDNA was primarily amplified from single molecules. By determining the endpoint dilution, we could quantify the number of AST in each aliquot and determine the fraction of the infected cells with AST by using the number of HIV-1 DNA molecules in replicate aliquots for the denominator. HIV-1 DNA was measured with the integrase cell-associated DNA (iCAD) assay (Hong et al, 2016). The number of HIV-1 DNA molecules is a surrogate for the number of infected cells since previous studies showed that most infected cells carry only a single provirus both before and during ART (Josefsson et al, 2013). For a subset of the samples, a 1.7-kb fragment of AST was also amplified at an endpoint and Sanger sequenced to assess the genetic diversity of the transcripts.

## Overview of the development and optimization of AST quantification and sequencing

During the development and optimization of the HIV-1 AST quantification and sequencing assays, we controlled for endogenous priming during cDNA synthesis and contamination of HIV-1 DNA. We also determined the sensitivity and background of the AST assays. An overview of these developmental steps is provided here, but detailed protocols for each set of experiments are provided below.

1. To eliminate endogenous self-priming during AST cDNA synthesis (Zapata et al, 2017; Mancarella et al, 2019), we used a 5′ end exogenous oligo-tagged gene specific primer (Table S1). The tag generates cDNA carrying the exogenous oligo-tag sequence to function as the target for the forward primer during first-round PCR amplification; thus, allowing for specific amplification of cDNA molecules containing the exogenous oligo-tag sequence, rather than endogenous self-primed templates.

2. To ensure complete DNA digestion, we used various amounts of the ACH-2 infected cell line (~1 provirus/cell), spiked into $1 × 10^5$ or $1 × 10^6$ uninfected CEM cells (Fig S2). Nucleic acid was extracted, DNase treated, and "cDNA synthesized" without reverse transcriptase. Nested PCR was performed to determine the maximum number of cells that can be assayed to ensure that HIV DNA digestion was complete. We found complete digestion of viral DNA at 100 infected cells per aliquot, in agreement with prior findings with the CARD-SGS assay where <300 infected cells per aliquot was the upper limit (Wiegand et al, 2017).

3. To optimize the AST assay, we used in vitro transcribed AST generated with a pMiniT-AST vector containing a T7 promoter (Fig S3A–D). The concentration of AST was measured by spectrophotometry and $10^4$ copies were spiked into nucleic acid extracted from $1 × 10^5$ uninfected CEM cells. The number of positives detected in the assay was compared with the known number of transcripts. The optimized AST detection assay had a sensitivity of ~50% (average of 5,589 copies detected), which could be a consequence of including the exogenous oligo-tag on the cDNA primer reducing cDNA synthesis efficiency or loss of RNA template from the purification steps of the assay.

4. After optimizing the assay using in vitro transcribed AST, we tested the assay on 100 ACH-2 cells spiked into $10^5$ uninfected CEM cells (Fig S3E and F). Using the AST assay (43 bp amplicon) at endpoint with probe detection, we found a median of 41 AST molecules per 100 infected ACH-2 cells [IQR 25–69] (n = 16 replicates). We detected 13 AST/100 infected ACH-2 cells when amplifying and sequencing a 1.7-kb fragment of anti-*env*. The less frequent detection of the 1.7-kb amplicon is not unexpected compared with the 43 bp amplicon.

5. To determine the cut-off (LOQ) for the digital (probe detection at an endpoint) AST assay, we performed replicates of "no reverse-transcriptase (RT)" controls on 100 ACH-2 cells spiked into $10^5$ uninfected CEM cells. We found four potential false positive signals in 650 infected cells assayed when RT was not included in the cDNA reaction, making our assay cut-off 0.6 AST/100 infected cells. We also tested $5 × 10^5$ uninfected CEM cells spread across 182 PCR wells and detected 1 false positive, making the false-positive assay background <0.0002 copies/100 cells assayed. To overcome the background, we limited each digital assay to <100 infected cells and included equal numbers of "no RT" controls on each PCR plate.

## Cell lines, maintenance, and storage

Cell lines used for assay development were uninfected CEM/C1 cells (#CRL-2265; American Type Culture Collection) and HIV-1 infected ACH-2 cells (#ARP-349; HIV Reagent Program). ACH-2 cells are A3.01 cells infected with a single integrated provirus (HIV-1$_{LAI}$). Complete medium was prepared as RPMI 1640 supplemented with 1% Penicillin-Streptomycin-Glutamine (#10378016; Gibco) and 10% heat-inactivated fetal bovine serum (HI-FBS). Cell Freeze Medium was prepared as RPMI 1640 supplemented with 10% HI-FBS (CEM/C1 cells) or 10–40% HI-FBS (ACH-2 cells), and 10% DMSO.

Cells were maintained in complete medium (as described above) and seeded in 2x vented T-25 flasks (1:3 and 2:3 from vial contents) with a total of 6 ml in each flask. Cells were incubated overnight at 37°C, 5% $CO_2$ with flasks upright. The following day, the medium was aspirated without disturbing the cells and were replenished with warmed complete medium. The cells were subcultured and transferred to a 15 ml conical tube for centrifugation at 150$g$ for 10 min, then media was aspirated. The cells were resuspended in 1 ml complete medium and counted by hemocytometer at 1:10 and

1:100 dilutions. Viable cells were seeded at 2–4 × $10^5$ in new T-25 flasks with 6 ml of warmed complete medium. If needed, cells were expanded in T-75 flasks (seeding at 4–6 × $10^5$) with medium volume between 10–20 ml. The cells were split every 2–3 d as needed.

The cells were prepared in Cell freeze medium at a minimum of 1 × $10^6$ cells/ml by gentle pipetting of cells and aliquoted into labeled cryotubes. They were placed in a freezing container with 100% isopropanol in outer container to freeze cells at –80°C overnight. The following day, the cells were transferred to $LN_2$ storage.

### Sample preparation

To thaw viably frozen cells, RPMI 1640 was warmed to 37°C and added dropwise to thaw PBMC for nucleic acid extraction. Each viably frozen cell vial was warmed for ~2 min at 37°C prior to RPMI being added dropwise to the vial. Thawed donor PBMC was aliquoted and centrifuged at 500$g$ for 5 min. Supernatant was removed and the pelleted cells were used immediately for downstream assays or were frozen on dry ice and stored back in $LN_2$ until ready for use.

### Generation of control RNA

RNA controls were generated to determine the efficiency of the HIV-1 AST cDNA synthesis and PCR amplification. AST (*anti-env*) were amplified from a pUC57-AST plasmid obtained from Fabio Romerio (Department of Molecular and Comparative Pathobiology, The Johns Hopkins University School of Medicine, Baltimore, MD) using M13 Fwd and M13 Rev primers (Table S2). Using Platinum II Taq (#14966025; Thermo Fisher Scientific), 50 µl PCR reactions were carried out with 50 ng of plasmid DNA and 10 µl of 5X Platinum II PCR Buffer, 1 µl of 10 mM dNTPs, 1 µl of each 10 µM primer, 0.4 µl of Platinum II Taq polymerase, and molecular-grade water to bring to 50 µl. PCR cycling was performed as follows: 94°C for 2 min and 45 cycles of 94°C for 15 s, 60°C for 15 s, and 68°C for 45 s, followed by a final extension at 68°C for 1 min. The PCR product was then purified using a QIAquick PCR Purification Kit (#28104; QIAGEN). The PCR product was cloned using NEB PCR Cloning Kit (#E1202S; New England Biolabs) where 80 ng of the purified PCR product was ligated into a linearized pMiniT 2.0 vector and transformed into the NEB stable competent 10-beta *E. coli* following manufacturer's instructions (#C3019H; New England Biolabs). Transformed colonies were picked, grown overnight according to manufacturer's instructions, and miniprepped using the QuickLyse Miniprep Kit (#27406; QIAGEN). Plasmids were screened for the insert by selective PCR screening using Cloning Analysis Fwd and Rev primers (Table S2) with PCR cycling performed as follows: 94°C for 2 min and 45 cycles of 94°C for 15 s, 60°C for 15 s, and 68°C for 45 s, followed by a final extension at 68°C for 1 min. The plasmid was then Sanger sequenced to confirm orientation and sequence (Table S2).

To generate control transcripts, 1 µg of the AST clone (mentioned above) was digested by restriction enzyme PmeI (#R0560S; New England Biolabs) for AST (*anti-env*) or ZraI (#R0659S; New England Biolabs) for *env* transcripts following manufacturer's instructions. Digested DNA was then purified using the QIAquick PCR Purification Kit (#28104; QIAGEN) and concentrated using ethanol precipitation

(described below). RNA was transcribed using HiScribe T7 Quick High Yield RNA Synthesis Kit (#E2050S; New England Biolabs) for AST or the HiScribe SP6 Quick High Yield RNA Synthesis Kit (#E2070S; New England Biolabs) for *env* transcripts. Transcribed RNA was purified using a Monarch RNA Cleanup Kit (#T2030; New England Biolabs) and quantified by NanoDrop spectrophotometer. The RNA was diluted to 1 × $10^6$ copies/ml with 5 mM Tris–HCl pH 8.0 and stored at –80°C for temporary storage and $LN_2$ for longer storage to help avoid degradation of RNA.

To assess the completeness of digestion of plasmid DNA, ~$10^4$ copies of the transcripts were spiked into extracted nucleic acid from $10^5$ CEM cells and DNase I treated (described below). The control transcripts were used for cDNA synthesis using random hexamer primers (#SO142; Thermo Fisher Scientific) in the following reaction: 2.5 µl 10 mM dNTPs and 2.5 µl 2 µM random hexamer primer were added to each 20 µl RNA sample. The RNA was denatured at 65°C for 10 min, then immediately cooled at –20°C for 1 min. Next, 25 µl of SuperScript III Reverse Transcriptase (#18080-044; Invitrogen) master mix was added to the denatured RNA. Master mix was made by combining 10 µl 5X first strand buffer, 1 µl 0.1 M DTT, 12 µl RNase-free water, 1 µl 40 U/µl RNaseOUT recombinant ribonuclease inhibitor (#10777-019; Invitrogen), and 1 µl 200 U/µl SuperScript III Reverse Transcriptase. cDNA was synthesized at 25°C for 5 min, 55°C for 1 h, 70°C for 15 min, then cooled to 4°C. To remove hybridized RNA, 1 µl RNase H (5 units) (#M0297S; New England Biolabs) was added and incubated at 37°C for 20 min followed by an enzyme heat-inactivation at 75°C for 10 min, and the cDNA was cooled to 4°C and used immediately or stored at –80°C.

### DNA and RNA extraction

Total nucleic acid was extracted from endpoint diluted aliquots of HIV-infected cells with AST RNA, determined by performing serial dilutions of PBMC until 96-well PCR plates yielded <30% AST positives. Nucleic acid was extracted by adding 100 µl 3 M guanidine HCl solution (18.75 ml 8 M guanidinium HCl, 2.5 ml 1 M Tris–HCl pH 8.0, 0.5 ml 100 mM $CaCl_2$) and 5 µl 20 mg/ml Proteinase K to a cell pellet, vortexed, and incubated at 42°C heat block for 1 h. Addition of 400 µl 6 M guanidine isothiocyanate solution (25 ml 6 M guanidinium isothiocyanate, 14 ml 1 M Tris–HCl pH 8.0, 53 µl EDTA pH 8.0) and 6 µl 20 mg/ml glycogen to lysed cell pellet was vortexed, and incubated at 42°C heat block for 10 min. Five hundred microliters of 100% isopropanol was added with mixing, followed by centrifugation at 21,000$g$ for 10 min at RT to precipitate nucleic acids. Supernatant was removed without disturbing the pellet and washed with 750 µl 70% ethanol. Precipitated nucleic acid was air-dried. For DNA, the pellet was resuspended in 100 µl 5 mM Tris–HCl pH 8.0 and ready for use. For RNA, the DNA was digested by either (i) DNase I (#04716728001; Roche) or (ii) ezDNase (#11766051; Thermo Fisher Scientific):

(i) DNase I: The air-dried extracted nucleic acid was resuspended in 38 µl DNase buffer and 2 µl of 10 units/µl DNase I and incubated for 20 min in a 37°C water bath. After incubation, 200 µl of 6 M GuSCN (#50983; Sigma-Aldrich) was added and mixed well, followed by the addition of 250 µl of 100% isopropanol. Sample was vortexed for 10 s, then centrifuged at

21,000g for 10 min to pellet nucleic acid. Supernatant was removed and pellet was washed with 1 ml 70% ethanol, followed by additional centrifugation at 21,000g for 10 min. Finally, the DNA-digested RNA pellet was air-dried, resuspended in 20 µl of 5 mM Tris–HCl pH 8.0, and used immediately for cDNA synthesis. Alternatively, RNA could be temporarily stored in 70% ethanol at −80°C.

(ii) ezDNase: The air-dried extracted nucleic acid was resuspended in 32 µl 5 mM Tris–HCl pH 8.0. Sample was split in equal volumes to control for successful DNA digestion. To ensure successful DNA digestion, manufacturer's instructions were followed, with volumes of reagents doubled and maximum incubations used. Briefly, 2 µl 10X ezDNase Buffer and 2 µl ezDNase were added to each sample and incubated at 37°C for 5 min. To inactivate the ezDNase, the mixture was incubated at 55°C for 5 min in the presence of 2 µl 0.1 M DTT. Incubations were performed in a thermocycler. Samples were immediately cooled at −20°C and brought up to RT before directly moving into cDNA synthesis.

### cDNA synthesis

The cDNA was synthesized by adding 2.5 µl 10 mM dNTPs and 2.5 µl 2 µM gene-specific oligo-tagged cDNA primer (AST cDNA, Table S1) to each sample. The RNA was denatured at 85°C for 10 min, then immediately cooled at −20°C for 1 min. Next, 25 µl SuperScript III Reverse Transcriptase (#18080-044; Invitrogen) master mix was added to the denatured RNA. Master mix was 10 µl 5X first strand buffer, 0.5 µl 0.1 M DTT, 13.5 µl RNase-free water, 0.5 µl 40 U/µl RNaseOUT recombinant ribonuclease inhibitor (#10777-019; Invitrogen), and 1 µl 200 U/µl SuperScript III Reverse Transcriptase. cDNA was synthesized at 45°C for 1 h, then cooled to 4°C. To remove hybridized RNA, 1 µl RNase H (5 units) (#M0297S; New England Biolabs) was added and incubated at 37°C for 20 min followed by an enzyme heat-inactivation at 75°C for 10 min. The cDNA was cooled to 4°C and was transferred to a low bind Eppendorf tube and 0.1 vol 3 M sodium acetate pH 5.5 (#AM9740; Invitrogen) and 1 µl 20 mg/ml glycogen (#1090139300; Roche) were added. After mixing, three volumes 95% ethanol was added, and the entire mixture was vortexed for 10 s and incubated overnight at −20°C. The next day, the precipitated cDNA was centrifuged at 21,000g for 20 min at RT, and supernatant was removed. The pellet was washed with 600 µl 70% ethanol, pulse vortexed, and centrifuged for 15 min at 21,000g at RT. Supernatant was removed and cDNA pellet was air dried and then resuspended in 20 or 50 µl (for digital PCR quantitation) in 5 mM Tris–HCl pH 8.0. The cDNA was then used immediately or stored at −80°C.

### Quantification of HIV-1 infected cells

The number of HIV infected cells was estimated from the levels of HIV DNA in PBMC. For donors on ART, HIV-1 DNA levels were previously measured using the integrase cell-associated single-copy DNA (iCAD) assay (primers in Table S3) (Hong et al, 2016; McManus et al, 2019). We also measured HIV-1 DNA levels using a modified primer probe set in the Rev response element (RRE) adapted from the Intact Proviral DNA Assay (Bruner et al, 2019) (Table S4). Digital PCR was performed using Lightcycler 480 Probes Master hot-start reaction mix (#4707494001; Roche) in 20 µl volumes per reaction. A 2 ml master mix was prepared by combining 1 ml 2X Lightcycler 480 Probes Master hot-start reaction mix, 12 µl 100 µM RRE Fwd primer, 12 µl 100 µM RRE Rev, 2 µl 100 µM RRE probe, 774 µl RNase-free water, and 200 µl diluted DNA (DNA was diluted up to 200 µl with 5 mM Tris–HCl pH 8.0). Digital PCR was performed on the Roche Light-Cycler480 at a denaturation temperature of 95°C for 10 min and 55 cycles (95°C for 15 s and 60°C for 1 min). Equal numbers of "no RT" controls and experimental wells were included on each plate to ensure complete HIV-1 DNA digestion. Two negative (no template, 5 mM Tris–HCl pH 8.0) and two positive (100 ACH-2 in a background $1 \times 10^5$ CEM cells of DNA) controls were also included on each PCR plate. The number of positive wells was used to calculate the number of infected cells per million PBMC in each of the donor samples.

### Design of gene-specific primers and probes

Due to intra- and inter-host HIV-1 sequence variation, we extracted HIV-1 DNA from PBMC (as described above) and performed PCR at endpoint targeting *env* (primers in Table S1) and performed Sanger sequencing (primers in Table S4). The designed primers for were used accordingly (Table S5, donor-specific mutations are shown in bold red).

### AST quantification by digital PCR

AST digital PCR was performed using Lightcycler 480 Probes Master hot-start reaction mix (#4707494001; Roche) in 20 µl volumes per reaction. A master mix was prepared by combining 2X Lightcycler 480 Probes Master hot-start reaction mix, 100 µM Fwd primer, 100 µM Rev primer, 100 µM AST probe, RNase-free water, and cDNA synthesized as described above. Digital PCR was performed on the Roche LightCycler480 at a denaturation temperature of 95°C for 10 min and 55 cycles (95°C for 15 s and 60°C for 1 min). Equal numbers of "no RT" controls and experimental wells were included on each plate to ensure complete HIV-1 DNA digestion. Two negative (no template, 5 mM Tris–HCl pH 8.0) and two positive (100 ACH-2 in a background $1 \times 10^5$ CEM cells of DNA) controls were also included on each PCR plate.

### HIV-1 *env* transcript quantification by digital PCR

The same nucleic acid extraction method as AST was used and then followed with DNase treatment. Conditions for cDNA synthesis were the same except with a different strand-specific oligo-tagged cDNA primer (Env cDNA, Table S1). Conditions for *env* transcript quantification by digital PCR were the same as AST by except for the primers and probe (*env*, Table S3).

### AST quantification and sequencing

AST quantification and sequencing was modified from the original CARD-SGS assay protocol as previously described (Wiegand et al, 2017; Capoferri et al, 2024) and was used to sequence and quantify HIV-1 DNA and sense and antisense HIV RNA in *gag*, *pol*, and *env*.

Briefly, cDNA or DNA was diluted to a near endpoint in a volume of 200 μl and added to 800 μl Platinum II Taq (#14966025; Thermo Fisher Scientific) master mix containing 200 μl 5X Platinum II PCR Buffer, 20 μl 10 mM dNTPs, 8 μl primers (4 μl 50 μM Forward and 4 μl 50 μM Reverse primers, for respective region, Table S1), 564 μl molecular-grade water, and 8 μl Platinum II Taq polymerase. The total volume of cDNA and master mix was spread across a 96-well plate (10 μl per well). PCR cycling was performed with respect to sub-genomic region (Table S1). Following the first round of PCR, each well was diluted with 50 μl 5 mM Tris–HCl pH 8.0. For nested PCR, 2 μl diluted PCR 1 product was transferred from each well to a 96-well plate containing 8 μl Platinum II Taq master mix (described above). Nested PCR was performed with respect to sub-genomic region with respective thermocycling conditions (Table S1). Nested PCR product was diluted with 50 μl 5 mM Tris–HCl pH 8.0. Positive wells were identified by GelRed detection (#41003; Biotium) using UV imaging. Positive PCR reactions were sent for Sanger sequencing (Table S4).

The levels of HIV expression in single cells were determined by the number of identical sequences in each aliquot. Because the reverse transcription step is known to introduce errors at a rate of about $10^{-4}$ per sequenced nucleotide of viral cDNA sequences (Mansky & Temin, 1995), RT–PCR variants that differed by a single nucleotide ("fuzz") from a group of seven or more identical sequences within the same aliquot were, conservatively, considered to belong to the same rake of identical sequences. The R script for "defuzzing" is available at https://github.com/michaelbale/RStuff/blob/master/fPECS_fncs.R.

### Sequence analysis

Sequences were aligned with MAFFT v7.450 (FFT-NS-1 200PAM/k = 2 algorithm) in Geneious Prime 2020.2.4. Minor adjustments were performed manually. P-distance neighbor joining trees were reconstructed using MEGA X and rooted on the consensus B HIV sequence. Population genetic diversity was calculated as APD using MEGA 11. APOBEC3-G/F mediated hypermutation was predicted using the Los Alamos National Laboratory HIV Sequence database tool, Hypermut (https://www.hiv.lanl.gov/content/sequence/HYPERMUT/hypermut.html).

### Statistical analyses

Wilcoxon rank-sum test, Wilcoxon signed-rank paired test, Kruskal-Wallis test, Mann–Whitney *U* test, Spearman's correlation, and Shapiro-Wilk test for normality were performed using Prism GraphPad 9.2. or in R (version 4.3.1). Spearman's correlation was selected due to the random variation in biological systems.

To test if there was a difference in the levels of AST pre- and post-ART interruption we used a binomial test. From the two sample donations pre-interruption, a total of 155 infected PBMC were assayed and 39 infected PBMC were found to contain AST (25%). Post-interruption, 133 infected PBMC were assayed and 65 infected PBMC contained AST (49%). A Binomial distribution was used to show that 65 out of 133 is significantly larger than 39 out of 155.

Statistical tests applied are indicated in the text and/or figures/table legends. Additional statistical analyses were performed using R (version 4.3.1).

## Data and Material Availability

Data are available in the article itself and its supplementary materials. All sequence data are available in a public repository at GenBank (PQ612087–PQ612412). No new HIV-1 DNA sequence data were generated from the donors on ART which were previously published by McManus et al (2019).

## Supplementary Information

## Acknowledgements

We would like to thank the donors for participating in this study. We acknowledge our collaborative interactions with the Behavior of HIV in Viral Environments Center (U54AI170855). We thank Teresa Burdette and Ann Wiegand for administrative support. This work was supported by intramural NCI funding (ZIA BC 011697) to the HIV Dynamics and Replication Program (MF Kearney) and by the Office of AIDS Research. This work was also supported through an Intramural AIDS Research Fellowship (AA Capoferri) from the Office of AIDS Research. Other funders include NCI contract No. 75N91024F00011 to BT Luke, Advanced Biomedical Computational Science subcontract 12XS547 to JW Mellors and 13SX110 to JM Coffin. JM Coffin was a Research Professor of the American Cancer Society and supported in part by Research Grants CA R35 200421 and AI R01 184043. The content is solely the responsibility of the authors and does not necessarily reflect the official views of the National Cancer Institute, the National Institutes of Health, or the Department of Health and Human Services.

### Author Contributions

AA Capoferri: conceptualization, data curation, formal analysis, validation, investigation, visualization, methodology, project administration, and writing—original draft, review, and editing.
R Sklutis: conceptualization, data curation, formal analysis, validation, investigation, visualization, methodology, project administration, and writing—original draft, review, and editing.
TO Famuyiwa: formal analysis, methodology, and writing—review and editing.
S Pathak: methodology and writing—review and editing.
R Li: resources, methodology, and writing—review and editing.
JW Rausch: conceptualization, formal analysis, supervision, visualization, and writing—review and editing.
BT Luke: formal analysis and writing—original draft.
R Hoh: resources and writing—review and editing.
SG Deeks: resources and writing—review and editing.
JW Mellors: resources and writing—review and editing.
JM Coffin: conceptualization and writing—review and editing.

JL Groebner: conceptualization, formal analysis, supervision, investigation, visualization, methodology, project administration, and writing—original draft, review, and editing.
F Romerio: conceptualization, resources, and writing—original draft, review, and editing.
MF Kearney: conceptualization, formal analysis, supervision, funding acquisition, visualization, and writing—original draft, review, and editing.

## Conflict of Interest Statement

JW Mellors is a consultant to Gilead Sciences, has received research grants from Gilead Sciences to the University of Pittsburgh, and owns share options in Infectious Disease Connect (co-founder) and Galapagos, NV, unrelated to the current work on HIV. JM Coffin is a member of the Scientific Advisory Board and a Shareholder of ROME Therapeutics, Inc. and Generate Biomedicine, Inc. The remaining authors have no potential conflicts.

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
