## [Reviewer comments · Life Science Alliance]

Life Science Alliance

In vivo detection of antisense HIV-1 transcripts in untreated and ART-treated individuals

Adam Capoferri, Rachel Sklutuis, Toluleke Famuyiwa, Sachi Pathak, Rui Li, Jason Rausch, Brian Luke, Rebecca Hoh, Steven Deeks, John Mellors, John Coffin, Jennifer Groebner, Fabio Romerio, and Mary Kearney

DOI: <https://doi.org/10.26508/lsa.202503204>

Corresponding author(s): Adam Capoferri, National Cancer Institute and Mary Kearney, National Cancer Institute

Review Timeline:

Submission Date:	2025-01-06
Editorial Decision:	2025-03-17
Revision Received:	2025-06-02
Editorial Decision:	2025-06-18
Revision Received:	2025-06-26
Accepted:	2025-06-27

Scientific Editor: Tim Fessenden

Transaction Report:

March 17, 2025

Re: Life Science Alliance manuscript #LSA-2025-03204-T

Dr. Adam A Capoferri
National Cancer Institute
535 SULTAN STREET RM 111
Frederick 21702

Dear Dr. Capoferri,

Thank you for submitting your manuscript entitled "In vivo detection of antisense HIV-1 transcripts in untreated and ART-treated individuals" to Life Science Alliance. The manuscript was assessed by expert reviewers, whose comments are appended to this letter. We invite you to submit a revised manuscript addressing the Reviewer comments.

Thank you for this interesting contribution to Life Science Alliance. We are looking forward to receiving your revised manuscript.

Sincerely,

B. MANUSCRIPT ORGANIZATION AND FORMATTING:

Reviewer #1 (Comments to the Authors (Required)):

In this study by Capoferri et al., the authors identified and quantified antisense transcripts of HIV (AST) in clinical samples. While the impact of AST on HIV has been characterized in previous years by Romerio et al. (also a co-author), the presence and quantity of AST in clinical samples remain unknown. As described in the intro, antisense transcripts in another retrovirus HTLV is highly important for HTLV pathogenesis, while the clinical significance of AST in HIV remains unknown. The authors leveraged their expertise in careful and meticulous approaches quantifying and phylogenetic analysis HIV RNA and identified a relatively high frequency of HIV AST transcripts in the blood and lymph nodes, even when the HIV+ person is under suppressive ART. This is a new finding with both clinical relevance.

A few points should be addressed to strengthen this manuscript:

Major:

1. Comparing the sensitivity of AST transcripts versus env transcripts: "we optimized the AST assay on antisense RNA in the ACH-2 cell line and found these cells to express a median of 41 AST per 100 infected ACH-2 cells" - to make the downstream comparison even more rigorous, do the authors have the copy number (detection limit) of env RNA and long AST in this ACH-2 cell line?
2. Figure 1. Do untreated participants have higher AST:env ratio, like that seen in treated participants shown in Figure 1B? Normalization: can the authors add the frequency of AST and env (like Figure 1B) on untreated participants? Then, can the authors plot the ratio of AST:env (like a fold change) in both treated and untreated participants? Does AST expression increase after ART (seems to be what Figure 1A suggests) while env (sense) expression decreases? If the authors plot a correlation plot and examine with Spearman correlation, do the sense (env) and antisense (AST) copy number correlate positively or negatively, with and without ART (separate plots)?
3. Line 173: "median 26 copies/100 infected PBMC [IQR 16 - 47] vs. median 16 copies/100 infected PBMC [IQR 5 - 25]) (paired t test $t(5)=2.79$, $p=0.04$)". Note that median/IQR are used for non-parametric data analysis while t-test is for parametric data analysis (for data following Gaussian distribution). Since the detection has cut-offs (detection limit) and does not follow Gaussian distribution, please use a nonparametric test (eg. Wilcoxon, paired) for this analysis. Is the p value still statistically significant?
4. Table 2 and interpretations (lines 184-186, lines 238-240) are incorrect. Given that infected cells have unequal number of HIV RNA expressed, reporting an average is wrong. Basically, HIV RNA expression (burst size) is exponentially different between cells, and an average (assuming Gaussian distribution) cannot be used. For example, in one scenario, one infected cell expresses 1000 HIV RNA while the remaining 99 infected cells express 0 HIV RNA. In another scenario, 100 infected cells each express 10 copies of HIV RNA, as demonstrated by studies from Frank Maldarelli and Mathias Lichterfeld. Thus, the authors should report what was exactly measured, such as x copies per 100 PBMC, not trying to average out the number as if HIV RNA copy numbers are equal. Similarly, please remove "% expressing" in Figure 3, unless it's performed using simultaneous DNA/RNA detection in the same 96 well limiting dilution (as done by Patro et al. PNAS 2019).
5. Line 186-187: "indicating that the frequency of detection of "long AST" was lower than the short AST fragments detected by the digital PCR approach (above)". What's the detection limit of long versus short AST in ACH-2 cells? Is this difference due to technical difference (different sensitivity of assays of detection, assuming that long forms are always less efficient in detection than short forms), or is it a true biological finding?

Minor:

1. Figure 1A and line 153: please label 0.048 directly
2. Figure 3 and Figure S1: HIV protein names (eg. Gag Pol, illustrated as different frames) should have first letter capitalized.

Reviewer #2 (Comments to the Authors (Required)):

The manuscript entitled "In vivo detection of antisense HIV-1 transcripts in untreated and ART-treated individuals" reported detection and quantification of antisense HIV-1 transcripts (AST) in people with HIV (PWH) using digital PCR and a modified CARD-SGS assay. The experiments were performed with proper validation and quality controls. However, the results did not support some of the conclusions and the authors did not provide enough evidence to support the importance of detecting AST in vivo. Please see below for the specific comments.

1. The authors stated that "expression levels of AST may be higher during ART compared to untreated individuals". First, they did Mann-Whitney test on data that contain data points from the same individual in one group (Fig. 1A), which violated the assumption of the test: observations within each group are independent of one another. The same error applies to paired T test

used in Fig. 1B. The authors also stated that "we found 13-fold higher levels of AST in PBMC samples collected on ART compared to those not on ART". I am not sure how the four data point from the same person affects this fold change as after they combined the data points from that individual the difference was no longer significant. Then the authors used Grubbs' Test for Outliers to determine that one out of 5 levels of AST in untreated PWH was an outlier. This test is improperly used. The assumption of the test is that the data are normally distributed and the sample size is too small to determine if the assumption is fulfilled. The authors need to increase sample size to test if expression levels of AST differed by ART status of PWH.

2. The authors stated that "We detected HIV-1 AST in all 8 donors independent of treatment status". In one donor, only 1 AST molecule was detected, which was below the cut-off of the assay (Fig 1b). As the cut-off of the assay was determined by false-positive signals in assays without reverse transcriptase and false-positive signal in uninfected cells, can the author determine whether the 1 AST molecule in this donor is false-positive or not?

3. The authors only examined the levels of ATS in one donor longitudinally before ART interruption and after ART initiation. It is not appropriate to overinterpret the data and suggest that treatment interruption may influence AST expression. With only one person examined, it is impossible to determine if the observed changes of ATS expression was due to ART interruption or the natural variance of ATS in this person.

4. The percentage of HIV-infected cells containing AST is very low (median: 4.1%). Even after accounting for the poor recovery of AST as discussed by the authors, the results did not support that AST may determine HIV persistence.

5. Only a few ATS sequences matched with the sequences of HIV-provirus (Fig. 3). Can authors speculate how the other non-matching ATS were generated in the cells and how they would promote HIV latency?

6. The authors stated that levels of HIV-1 ATS in this study were comparable to those reported levels of some eukaryotic AST. The authors also mentioned some ATS were expressed at high levels. Do the authors know if the function of ATS could be affected by their levels of expression? In addition, it would be informative to compare the levels of HIV-1 ATS with levels of ATS encoded by other Retroviridae families.

On behalf of my co-authors, I would like to thank the Reviewers for their careful reading of our manuscript. We revised the manuscript to address your concerns and performed additional experiments as recommended. The Reviewers' insights have improved our study and manuscript. Below, we provide point-by-point responses to each comment. Changes to the manuscript are shown in red.

Reviewer #1 (Comments to the Authors (Required)):

In this study by Capoferri et al., the authors identified and quantified antisense transcripts of HIV (AST) in clinical samples. While the impact of AST on HIV has been characterized in previous years by Romero et al. (also a co-author), the presence and quantity of AST in clinical samples remain unknown. As described in the intro, antisense transcripts in another retrovirus HTLV is highly important for HTLV pathogenesis, while the clinical significance of AST in HIV remains unknown. The authors leveraged their expertise in careful and meticulous approaches quantifying and phylogenetic analysis HIV RNA and identified a relatively high frequency of HIV AST transcripts in the blood and lymph nodes, even when the HIV+ person is under suppressive ART. This is a new finding with both clinical relevance.

A few points should be addressed to strengthen this manuscript:

Major:

1. Comparing the sensitivity of AST transcripts versus env transcripts: "we optimized the AST assay on antisense RNA in the ACH-2 cell line and found these cells to express a median of 41 AST per 100 infected ACH-2 cells" - to make the downstream comparison even more rigorous, do the authors have the copy number (detection limit) of env RNA and long AST in this ACH-2 cell line?

We thank the reviewer for this question. Our prior work (Wiegand *et al.* PNAS 2017, PMID: 28416661) measured sense unspliced HIV RNA in unstimulated ACH-2 cell and found 715 copies per 100 infected cells, much higher than we detected for AST. The number of copies of sense HIV RNA ranged from 0 copies per cell to 40 copies per cell (range shown below in Fig S1 from Wiegand *et al.* PNAS 2017). Each 96-well plate below is endpoint-diluted HIV unspliced RNA from one unstimulated ACH2 cell. The number of sense RNA (cDNA) molecules detected is indicated above each plate. We have revised our manuscript on pages 6-7 to include this comparison to AST and referenced the paper and figure showing these data.

Fig S1. PCR results from CARD-SGS performed on 20 replicates of an average of one ACH-2 cell spiked into 500,000 PBMCs from an HIV-negative donor. The highlighted wells contain

amplification products from one or more HIV cDNA molecules. The dark wells are negative for HIV cDNA. The number above each plate shows the number of single HIV RNA molecules detected in each aliquot of spiked cells.

2. Figure 1. Do untreated participants have higher AST:env ratio, like that seen in treated participants shown in Figure 1B? Normalization: can the authors add the frequency of AST and env (like Figure 1B) on untreated participants? Then, can the authors plot the ratio of AST:env (like a fold change) in both treated and untreated participants? Does AST expression increase after ART (seems to be what Figure 1A suggests) while env (sense) expression decreases? If the authors plot a correlation plot and examine with Spearman correlation, do the sense (env) and antisense (AST) copy number correlate positively or negatively, with and without ART (separate plots)?

We thank the Reviewer for suggesting these additional analyses. We included the additions in Figure 1 and modified the results and discussion on pages 5-6. We believe that these changes have improved our study and manuscript. Below is a summary of our modifications to Figure 1:

In Fig.1A, we compared AST levels between treated and untreated donors. Rather than using all timepoints for the statistics, we aggregated the data from multiple timepoints from PID 2669 (Timepoint 1&2, while they were on ART prior to ART-interruption) as suggested by Reviewer 2. However, in the text of the manuscript we also report value without aggregating the timepoints (without p-value due to violation of assumption that observations within each group are independent of one another) to drive the hypothesis that there may be differences in levels of AST between ART-treated and untreated PWH, with the hope of encouraging larger studies in the field.

In Fig 1B, we added a comparison of *env* transcript levels in treated and untreated donors, which were not significantly different at our level of sampling. In one donor, PID 1508, we did not detect *env* transcripts, thus, we took an upper limit approximation of 1/number of infected cells assayed (open shape). The failure to detect a difference in the levels of HIV *env* expression in the ART treated vs. untreated group may be due to the small N in our study, the failure of PBMCs to contribute significantly to plasma viremia (compared to lymph nodes, for example), and/or *env* mismatches to the primers or probe in some cases. Although attempts were made to match primers to the population sequence within each donor, the high levels of HIV genetic diversity cannot guarantee matches to all variants within a population. In Capoferri *et al.* PNAS 2024 (PMID: 39190360), we showed that about 80-90% of infected PBMCs in untreated PWH do not express HIV RNA at a given point in time. Our finding here are consistent with our previous study.

In Fig 1C, we show the ratio of AST:*env* transcripts for each participant. We found a trend towards higher AST:*env* in the treated group. In the presence of ART, AST levels are higher than *env* transcript levels. Without ART, AST levels are lower than *env* transcript levels, but it does not reach significance in this small study. We hope that our small pilot study will encourage funding for large clinical studies to investigate the possible role of AST in HIV expression and persistence.

Fig.1D&E we show the comparisons of AST:*env* levels for each donor with an inset of the association. Figure 1D was originally Figure 1B upon initial submission where we found significantly more AST than sense HIV RNA ($p=0.03$, Wilcoxon signed-rank paired test) with a significantly positive association (Spearman $\rho=0.89$). Now, we added the same analyses of the untreated donors. In the untreated group, we did not find a significant difference in AST vs.

env transcripts ($p=0.19$, Wilcoxon signed-rank paired test) and no significant correlation (Spearman $\rho=0.21$).

It is important to note that, due to our use of oligo-tagged strand-specific primers for cDNA synthesis to prevent endogenous priming, we are unable to compare levels of AST and *env* transcripts in the same infected cell. The levels reported in Figure 1 reflect their bulk levels in replicate aliquots of PBMCs.

In summary, our new Figure 1 drive the hypothesis that HIV AST levels could be higher in PWH on ART, than in those who are not on ART, that the ratio of AST:*env* on ART is >1 , while it is <1 in those who are not on ART, and that there is a significant positive correlation between levels of AST and *env* transcripts on ART that is not present in PWH who are not on ART. These findings have been added to the Results pages 5-6, and the Discussion page 9.

3. Line 173: "median 26 copies/100 infected PBMC [IQR 16 - 47] vs. median 16 copies/100 infected PBMC [IQR 5 - 25]) (paired t test $t(5)=2.79$, $p=0.04$)". Note that median/IQR are used for non-parametric data analysis while t-test is for parametric data analysis (for data following Gaussian distribution). Since the detection has cut-offs (detection limit) and does not follow Gaussian distribution, please use a nonparametric test (eg. Wilcox, paired) for this analysis. Is the p value still statistically significant?

Thank you for bringing this error to our attention. We have replaced the paired t-test with the Wilcoxon signed-rank paired test. The p-value using the Wilcoxon signed-rank paired test is 0.03. We have corrected this error in the manuscript on page 6.

4. Table 2 and interpretations (lines 184-186, lines 238-240) are incorrect. Given that infected cells have unequal number of HIV RNA expressed, reporting an average is wrong. Basically, HIV RNA expression (burst size) is exponentially different between cells, and an average (assuming Gaussian distribution) cannot be used. For example, in one scenario, one infected cell expresses 1000 HIV RNA while the remaining 99 infected cells express 0 HIV RNA. In another scenario, 100 infected cells each express 10 copies of HIV RNA, as demonstrated by studies from Frank Maldarelli and Mathias Lichterfeld. Thus, the authors should report what was exactly measured, such as x copies per 100 PBMC, not trying to average out the number as if HIV RNA copy numbers are equal. Similarly, please remove "% expressing" in Figure 3, unless it's performed using simultaneous DNA/RNA detection in the same 96 well limiting dilution (as done by Patro et al. PNAS 2019).

We thank the Reviewer for this point, and we agree that infected cells have very different levels of HIV RNA. We changed "average" to "median" and included the range of AST levels across the single cells that we analyzed. We found that, of the single cells that contained AST, levels ranged from 1 copy (most cells with AST had only one copy) to as many as 30 copies in one single cell. We modified Table 2 and the text of the paper on page 6. The assay used for Table 2, different from the ddPCR assay, does not measure AST levels in bulk, but measures the fraction of the infected cells with AST and the levels of AST in single infected cells. The method was modified from the CARD-SGS assay which was published by Wiegand *et al.* PNAS 2017. We modified the text of the paper to make this approach more clear.

5. Line 186-187: "indicating that the frequency of detection of "long AST" was lower than the short AST fragments detected by the digital PCR approach (above)". What's the detection limit of long versus short AST in ACH-2 cells? Is this difference due to technical difference (different

sensitivity of assays of detection, assuming that long forms are always less efficient in detection than short forms), or is it a true biological finding?

We thank the Reviewer for this comment. The difference is likely both technical and biological. The sensitivity between CARD-SGS (long fragments) vs. qPCR (short fragments) was previously explored in Wiegand *et al.* PNAS 2017 (PMID: 28416661) where there was an average 5-fold difference (Wiegand *et al.* PNAS 2017, Table 2 below), with shorter fragments sometimes being detected at higher levels than longer fragments. It is possible that the differences may be due to degradation of RNA within cells (biological) and/or varying cDNA or PCR efficiencies. We have modified the discussion of the paper on page 11 to discuss these possibilities.

Table 2. Varying fraction of HIV-expressing cells per donor

PID	Years on cART	No. of PBMC genomes recovered	No. of infected cells analyzed	Estimated no. of infected cells expressing HIV RNA	Infected PBMCs expressing HIV RNA, %	Average no. of HIV RNA molecules detected per aliquot	Fold difference in HIV RNA copies compared with qPCR
001	0	270,000	136	5	4	19	10 (qPCR higher)
002	2	350,000	79	14	18	13	2 (CARD-SGS higher)
003	>4	400,000	82	5	6	9	2 (CARD-SGS higher)
		230,000	51	5	10	9	3 (CARD-SGS higher)
004	9	1,110,000	1,092	17	2	20	13 (qPCR higher)
		280,000	186	8	4	13	3 (qPCR higher)
Average		440,000	271	9	7	14	5.5

Minor:

1. Figure 1A and line 153: please label 0.048 directly.

Corrected

2. Figure 3 and Figure S1: HIV protein names (eg. Gag Pol, illustrated as different frames) should have first letter capitalized.

Corrected

Reviewer #2 (Comments to the Authors (Required)):

The manuscript entitled "In vivo detection of antisense HIV-1 transcripts in untreated and ART-treated individuals" reported detection and quantification of antisense HIV-1 transcripts (AST) in people with HIV (PWH) using digital PCR and a modified CARD-SGS assay. The experiments were performed with proper validation and quality controls. However, the results did not support some of the conclusions and the authors did not provide enough evidence to support the importance of detecting AST in vivo. Please see below for the specific comments.

1. The authors stated that "expression levels of AST may be higher during ART compared to untreated individuals". First, they did Mann-Whitney test on data that contain data points from the same individual in one group (Fig. 1A), which violated the assumption of the test: observations within each group are independent of one another. The same error applies to paired T test used in Fig. 1B. The authors also stated that "we found 13-fold higher levels of AST in PBMC samples collected on ART compared to those not on ART". I am not sure how the four data point from the same person affects this fold change as after they combined the data points from that individual the difference was no longer significant. Then the authors used Grubbs' Test for Outliers to determine that one out of 5 levels of AST in untreated PWH was an

outlier. This test is improperly used. The assumption of the test is that the data are normally distributed and the sample size is too small to determine if the assumption is fulfilled. The authors need to increase sample size to test if expression levels of AST differed by ART status of PWH.

We thank the Reviewer for these comments. We agree that larger studies are needed to determine if there is an association between levels of AST and ART status and we have softened the language in the abstract on page 2, the results on page 4, the discussion on pages 10-11), and we corrected the statistics in Figure 1. Our small pilot study can only drive the hypothesis that AST may have a biological role *in vivo* to encourage funding for larger studies in the field to investigate a possible correlation. However, this study is the first report of sequence evidence of AST in PWH, the first to show the fraction of infected cells that contain AST, and the first to measure AST levels within single cells. We feel that these findings contribute to the literature on AST and will encourage funding for larger trials in this understudied area of research.

Changes to the statistics were made in Figure 1 with an emphasis on using only one timepoint from donor 2669 (panels A-C) to make no claims of significance, only to drive a hypothesis worth testing in a large study. We also eliminated the text describing the single infected cell that had 30 copies of AST as an outlier.

2. The authors stated that "We detected HIV-1 AST in all 8 donors independent of treatment status". In one donor, only 1 AST molecule was detected, which was below the cut-off of the assay (Fig 1b). As the cut-off of the assay was determined by false-positive signals in assays without reverse transcriptase and false-positive signal in uninfected cells, can the author determine whether the 1 AST molecule in this donor is false-positive or not?

The Reviewer is correct here. In PID 3611, we did detect 1 positive well but it was not above the assay background. We have modified the manuscript to state that AST were detected in at least 7/8 donors on page 5 and on page 9.

3. The authors only examined the levels of AST in one donor longitudinally before ART interruption and after ART initiation. It is not appropriate to overinterpret the data and suggest that treatment interruption may influence AST expression. With only one person examined, it is impossible to determine if the observed changes of AST expression was due to ART interruption or the natural variance of AST in this person.

We have modified and softened the language in the Discussion on page 10. We additionally added, "While it is tempting to imagine that AST levels are influenced by ART duration and interruptions, it is not possible with our small sample size to know if these differences are a result of treatment status or only reflex the natural variation of AST expression *in vivo*". This case does serve as a case study to indicate that larger studies are needed to fully understand whether ART interruption plays a role in AST levels, and to then further explore the biological reasons. We hope that these observations will encourage others to investigate AST in case there is a real biological role that has been understudied and underappreciated.

4. The percentage of HIV-infected cells containing AST is very low (median: 4.1%). Even after accounting for the poor recovery of AST as discussed by the authors, the results did not support that AST may determine HIV persistence.

We agree that our data do not show a biological role for AST in HIV persistence, only show with sequence evidence that AST is expressed in a small number of cells at a given time *in vivo*. The biological role of these transcripts will have to be demonstrated to have (or not have) a biological role in HIV infection and persistence in future studies. Like sense HIV RNA, which is expressed in about 10-20% of HIV-infected cells at a given point in time in treated and untreated PWH (Capoferri *et al.* PNAS 2024), AST is expressed in a small number of infected cells. In this body of work, we refrained from language that suggests that AST may determine persistence, rather, we suggest that it should be explored as a potential contributor. Considering that most cells with AST have only a single detectable copy, it is likely that many cells that express AST will not be captured at a time in the lifecycle when it is detectable. This low expression level of AST in HIV-infected cells is consistent with antisense expression in HTLV-1 (Miura *et al.* PLoS Path 2019, PMID: 31738810) and in other organisms. We modified the paper to include this comparison in the discussion on page 10.

5. Only a few AST sequences matched with the sequences of HIV-provirus (Fig. 3). Can authors speculate how the other non-matching AST were generated in the cells and how they would promote HIV latency?

We think the Reviewer is referring to Figure 2 in this comment. The lack of matching proviral and AST sequences is due to sampling and the very high diversity of HIV. We sequenced proviral DNA and AST in replicate aliquots of infected cells. Therefore, matches would only be detected in large clones of infected cells, where some members of the clones were present in multiple aliquots of cells. Because the donors were in chronic infection, the HIV genetic diversity was very high, making each provirus different, except in large infected T cell clones. Where matches are not seen, the high HIV genetic diversity is the explanation. However, the fact that we found a high genetic diversity of the AST suggests that a large number of proviruses express AST, rather than just some cells in a few clones.

6. The authors stated that levels of HIV-1 AST in this study were comparable to those reported levels of some eukaryotic AST. The authors also mentioned some AST were expressed at high levels. Do the authors know if the function of AST could be affected by their levels of expression? In addition, it would be informative to compare the levels of HIV-1 AST with levels of AST encoded by other Retroviridae families.

This is a great question regarding whether the function of AST could be affected by the levels of expression. Work from Dr. Fabio Romerio (co-author, Li *et al.* Science Advances 2025, PMID: 40344061) suggests that when AST is ectopically expressed at high levels, latency reversal is either completely blocked or significantly reduced. And, when mutant forms of AST are ectopically expressed at high levels, the interaction with the 5' LTR or specific host factors is lost, and the ability of latency reversal is restored. However, these studies were not strictly *in vivo* but were either *ex vivo* or *in vitro*. The direct relationship between levels of AST and their function *in vivo* is not known. We hope that small pilot studies like ours will lead to larger studies that will ultimately determine whether or not AST has a biological role *in vivo*. If so, then, perhaps it can be exploited in strategies towards achieving HIV remission without ART.

Including a comparison of AST levels in HIV to other Retroviruses is an excellent suggestion. We modified the manuscript to include this comparison on page 10. The study by Miura *et al.* PLoS Path 2019 (PMID: 31738810) reported about 1-2 *Hbz* copies per HTLV-1+ cells, consistent with the low levels we observed with HIV AST.

June 18, 2025

RE: Life Science Alliance Manuscript #LSA-2025-03204-TR

Dr. Adam A Capoferri
National Cancer Institute
535 SULTAN STREET RM 111
Frederick 21702

Dear Dr. Capoferri,

Thank you for submitting your revised manuscript entitled "In vivo detection of antisense HIV-1 transcripts in untreated and ART-treated individuals". We would be happy to publish your paper in Life Science Alliance pending final revisions necessary to meet our formatting guidelines.

- Please be sure that the authorship listing and order is correct.
- Please add ORCID ID for secondary corresponding author - they should have received instructions on how to do so.
- Please add the X and Bluesky handles of your host institute/organization as well as your own or/and one of the authors in our system.
- The contribution selected for Rui Li does not qualify them for authorship. Please either update the contributions in our system and in the Author Contributions section of the manuscript, or let us know if the author needs to be removed (and added potentially to the acknowledgment section).
- Please move your main, supplementary figure, and table legends to the main manuscript text after the references section.
- We encourage you to revise the figure legends for Figure 2 such that the figure panels are introduced in alphabetical order.
- Please add a callout for Table S7 to your main manuscript text.

A. FINAL FILES:

B. MANUSCRIPT ORGANIZATION AND FORMATTING:

spreadsheets for the main figures of the manuscript. If you would like to add source data, we would welcome one PDF/Excel-file per figure for this information. These files will be linked as supplementary "Source Data" files.

Sincerely,

Reviewer #1 (Comments to the Authors (Required)):

The authors have addressed my questions sufficiently.

Reviewer #2 (Comments to the Authors (Required)):

The authors have fully addressed my comments. I have no more comments.

June 27, 2025

RE: Life Science Alliance Manuscript #LSA-2025-03204-TRR

Dr. Adam A Capoferri
National Cancer Institute
535 SULTAN STREET RM 111
Frederick 21702

Dear Dr. Capoferri,

Thank you for submitting your Research Article entitled "In vivo detection of antisense HIV-1 transcripts in untreated and ART-treated individuals". It is a pleasure to let you know that your manuscript is now accepted for publication in Life Science Alliance. Congratulations on this interesting work.

DISTRIBUTION OF MATERIALS:

Again, congratulations on a very nice paper. I hope you found the review process to be constructive and are pleased with how the manuscript was handled editorially. We look forward to future exciting submissions from your lab.

Sincerely,
